# Epigenetics in the Uterine Environment: How Maternal Diet and ART May Influence the Epigenome in the Offspring with Long-Term Health Consequences

**DOI:** 10.3390/genes13010031

**Published:** 2021-12-23

**Authors:** Irene Peral-Sanchez, Batoul Hojeij, Diego A. Ojeda, Régine P. M. Steegers-Theunissen, Sandrine Willaime-Morawek

**Affiliations:** 1Faculty of Medicine, University of Southampton, Southampton SO16 6YD, UK; daop1m14@southamptonalumni.ac.uk (D.A.O.); S.Willaime-Morawek@soton.ac.uk (S.W.-M.); 2Department Obstetrics and Gynecology, Erasmus MC, University Medical Center, 3000 CA Rotterdam, The Netherlands; b.hojeij@erasmusmc.nl (B.H.); r.steegers@erasmusmc.nl (R.P.M.S.-T.)

**Keywords:** DOHaD, maternal diet, epigenetics, embryo, preimplantation period, ART

## Abstract

The societal burden of non-communicable disease is closely linked with environmental exposures and lifestyle behaviours, including the adherence to a poor maternal diet from the earliest preimplantation period of the life course onwards. Epigenetic variations caused by a compromised maternal nutritional status can affect embryonic development. This review summarises the main epigenetic modifications in mammals, especially DNA methylation, histone modifications, and ncRNA. These epigenetic changes can compromise the health of the offspring later in life. We discuss different types of nutritional stressors in human and animal models, such as maternal undernutrition, seasonal diets, low-protein diet, high-fat diet, and synthetic folic acid supplement use, and how these nutritional exposures epigenetically affect target genes and their outcomes. In addition, we review the concept of thrifty genes during the preimplantation period, and some examples that relate to epigenetic change and diet. Finally, we discuss different examples of maternal diets, their effect on outcomes, and their relationship with assisted reproductive technology (ART), including their implications on epigenetic modifications.

## 1. Introduction

Non-communicable diseases (NCDs), such as diabetes, cardiovascular diseases (CVDs), neurological disorders, obesity, and some cancers, are on the rise, leading to around 34 million deaths worldwide per annum [1]. The significant increase in NCDs is closely linked to environmental exposures and lifestyle behaviours. It is also clear that exposure to these different largely modifiable conditions (i.e., poor diet, stress, and chemicals) from the preimplantation period onwards is important in the origin of these diseases [2]. The relationship between exposure to the maternal environment and the development and health of the foetus is captured in the developmental origins of health and disease (DOHaD) hypothesis [3]. 

The research in DOHaD, which has increased substantially from the 1990s, has shown that the maternal environment during pregnancy can affect programming during development and increase the risk of offspring developing long-term diseases [3,4]. Professor David Barker was one of the pioneer researchers to demonstrate the DOHaD phenomenon in his epidemiological studies in 1989, linking perinatal weight and the subsequent growth trajectory with health and disease risk in later life [5,6]. Since the development of the DOHaD hypothesis, many authors have linked birth weight with the development of NCDs. Initially, only low birth weight was related; however, some studies found that this relationship was U-shaped, indicating that high birth weight can also relate to the risk of developing pathologies associated with metabolic syndrome [7]. The maternal environment is one of the most studied conditions that can influence offspring phenotype. The maternal environment comprises both extrinsic and intrinsic factors. Extrinsic factors act from the outside and are related to the environmental conditions (pollution, exposure to chemicals, etc.), but intrinsic factors act from within an individual and are directly dependent on the mother’s condition (eating behaviours, metabolism, lifestyle, etc.). Despite the importance of human studies, animal studies provide an advantage in that environmental exposures can be controlled (extrinsic factors) and maternal and subject confounders can be reduced (intrinsic factors) [8]. These studies have demonstrated that the origin of NCDs can be attributable to the maternal and gestational environment, laying the foundation for DOHaD. These studies have further progressed to reveal that an array of environmental factors experienced during pregnancy, including toxins, stress, diet, and lifestyle, contribute to physiological and metabolic development, impacting foetal growth and offspring phenotype and health.

DOHaD research in nutritional studies has now found links between the quality of maternal diet during pregnancy and its possible contribution to changes in the genome expression of the offspring in utero. These changes act to coordinate the physiological, metabolic and growth characteristics, and are mediated through epigenetic mechanisms [9,10]. Epigenetic changes refer to different molecular states that affect the regulation of gene expression by changing the structural organisation of the DNA, but without changing the sequence [11]. Epigenetic changes during foetal development can depend on microenvironmental variations. Thus, environmental factors, such as maternal diet, during early pregnancy can affect maternal metabolism, which can also impact offspring development [12,13]. The main epigenetic changes include DNA methylation, histone modifications, and the expression of non-coding RNAs (ncRNAs). Whilst the association between the in utero environment and offspring health has been widely described over the years, the underlying molecular processes influencing the epigenome are generally unknown. The study of epigenetic changes that occur during pregnancy is not limited to natural conditions, but may also operate in clinical settings, such as ART. The next goal in DOHaD research will be to understand the molecular details linking the maternal and clinical environment to the epigenome, and the subsequent biological steps that lead to the onset of disease risk in later life [14]. 

This review will discuss the evidence on how maternal diet in mammalian models can impose epigenetic modifications in offspring, and can lead to the development of disease in adulthood. We will also review the thrifty gene concept, where dietary-mediated epigenetic changes cause downstream gene expression changes that directly control metabolism and homeostasis [15]. Finally, we will consider clinical ART conditions and their link with maternal diet in light of the preimplantation epigenetics underlying DOHaD. 

## 2. DOHaD and Epigenetic Modifications during Development

In the framework of the DOHaD hypothesis, epigenetic mechanisms have been considered as a possible precursor of changes in developmental programming [16]. In the early pregnancy periods, maternal malnutrition (overnutrition or undernutrition) can alter development throughout gestation [17], and one of the possible causes of these alterations could be linked to epigenetic changes. The intrauterine environment is susceptible to any small changes, which can induce responses in the offspring that could lead to a higher risk of developing NCD later in life [17,18].

Different studies pointed towards the possible idea that, across generations, a compromised diet can affect the phenotype and epigenome of the next generation, making them prone to developing metabolic [19] and even mental illnesses [20], by epigenetic variations that are subject to variations in diet [20]. These premises represent a starting point to understand how lifestyle and dietary behaviours during pregnancy can alter an offspring’s health by triggering epigenetic modifications during development. 

These epigenetic modifications will contribute to the expression, activity level, or silencing of a specific gene (examples below). These changes may be induced by means of adaptation to the environment and survival, as envisaged in the DOHaD hypothesis. Firstly, we will summarise the main types of epigenetic regulation of gene expression that occur during development.

### 2.1. DNA Methylation

One of the most common epigenetic modifications is DNA methylation. DNA methylation is of great importance in the regulation of genome functions and chromatin stability, among others [11].

DNA methylation is caused by the attachment of a methyl group to cytosine, which can generally occur when cytosine is located next to guanine at a cytosine–phosphate–guanine (CpG) site [21]. When there are high levels of DNA methylation at the CpG sites in the promoter regions of a gene, transcription factors are prevented from binding to the DNA, thus inhibiting the expression of that gene [21,22]. Conversely, when the CpG sites on CpG islands are unmethylated, gene expression is promoted [23,24].

In mammals, after fertilization, a general de-methylation process occurs, followed by de novo methylation as the parental genome becomes superseded by the new embryonic genome. The methylation of CpG sites during the first days after fertilization is catalysed by DNA methyltransferases (DNMTs),and is essential for tissue differentiation and maintenance during development [25,26].

DNA methylation has a crucial role in the development of the embryo and cell lineage specification, thereby regulating differential gene expression [22]. The pattern of DNA methylation in the periconceptional period may respond to environmental conditions to permit adaptations in development. This modification is well studied, and its understanding can elucidate how the environment and epigenetics in the early stages of development affect health in later life [22,27] (reviewed previously: [21,22,27,28]) (Table 1).

Some studies have focused on maternal exposure to, or ingestion of, toxic substances (tobacco, mercury, or arsenic), suggesting that, from conception, the embryo is susceptible to undergoing epigenetic changes (DNA methylation) when the maternal environment is compromised [56]. For example, exposure to arsenic during early pregnancy may be associated with changes in DNA methylation in the foetus, leading to reduced birth weight [56,57], and such exposures may increase the rate of miscarriage [58].

DNA methylation is crucial in the control of gene expression, as aberrant changes in DNA methylation patterns are associated with many diseases and cancers [59]. Alterations in DNA methylation patterns can be caused by genetic factors (polymorphism and mutations), or by factors such as diet, and the imbalance of essential nutrients in the metabolism of methyl groups [60].

### 2.2. Histone Modification

Histones are essential for DNA folding in eukaryotic cells, and also play a critical role in the regulation of gene expression. Histones are susceptible to different post-transcriptional modifications in the amino-terminal tail domain of histones, such as ubiquitination, phosphorylation, methylation, and acetylation, and these markers are often called histone codes [14]. 

Acetylation and methylation are the most common and most studied modifications in development [61]. Histone acetylation, through histone acetyltransferase activity, is related to gene expression and an open chromatin state, while deacetylation is related to gene silencing [62]. The methylation of histones by methyltransferases can promote or decrease gene transcription. Histone methylation has a crucial role in the assembly and compaction of heterochromatin, so when methylation increases the chemical attractions between DNA and histone tails, transcription decreases because the DNA compacts to nucleosomes and transcription-polymerase factors cannot access the DNA [63,64] (reviewed previously: [23,61,62,64,65]). 

Histone modification is an important means for regulating gene expression, as these modifications will affect the function and expression of proteins, which can affect the development of the embryos [23,66] (Table 1). Previous studies have found that histone modification can affect embryonic development, even to the point of embryonic lethality [67].

### 2.3. Non-Coding RNAs (miRNA, lncRNA)

Another epigenetic modification is non-coding RNA (ncRNA) [68,69]. ncRNAs can be classified according to their size, into small (sncRNAs) and long ncRNAs (lncRNAs). In mammals, sncRNAs [70] include microRNA (miRNA) [68] and Piwi RNA [69]. lncRNAs are involved in the emergence and development of germ cells and embryos [71]. lncRNAs are associated with different molecular processes, such as splicing, regulation of gene expression (transcriptional and post-transcriptional processes), and chromatin modifications [72] ]. Their mechanism of action is complex, and can involve the creation of repressive domains (of many kilobases) [73], and can be classified into the following three models: (i) competitor, competing with other molecules by binding to the DNA; (ii) recruiter/activator, related to epigenetic modifications by activating epigenetic modifiers (i.e., promoting DNA methylation); (iii) precursor, where lncRNA can be processed by RNase to produce shorter active RNAs (involved in post-transcriptional regulation) [74,75,76]. In mammals, lncRNAs also have an essential role in X chromosome inactivation [77] (reviewed previously: [77,78,79]).

Although ncRNAs are very important in embryological development and are known to be a key regulator in chromosome formation, their mechanism and alteration in relation to maternal diet is still unclear.

## 3. Diet Models and Epigenetic Modifications: Examples and Biological Meaning

Nutrition is an important player in gene regulation, as alterations in nutrition can influence gene expression, and, more importantly, it has been described that early alterations in maternal nutrition during development may be responsible for the development of chronic diseases through epigenetic mechanisms [80,81].

Human and animal studies of maternal nutrition during pregnancy, or just the preimplantation period, have shown that it affects the epigenome and offspring phenotype, and can be categorised into different models and studies, as outlined in the following sections. 

### 3.1. Dutch Famine

The effects of the Dutch famine of 1944–1945, during World War II, when Amsterdam was occupied for a five-month period by the Nazis, has been the subject of numerous epidemiological studies. During the famine, adult women received a daily food ration of some 400–800 calories, regardless of their social class. Immediately after the occupation, food became plentiful again, rations increased to 2000 calories, and, since medical records were detailed, the effects of malnutrition at key periods of pregnancy could be investigated [32,82,83]. Studies showed that men and women conceived during the famine, and who experienced famine during early pregnancy, were at an increased risk of developing metabolic diseases, such as obesity or diabetes II, in adulthood. Moreover, limited nutrient intake in early pregnancy decreased the development of pancreatic β-cells, resulting in glucose intolerance later in life [84,85]. 

Thus, among other modifications, a reduction was found in DNA methylation in famine-exposed offspring compared to same-sex controls, on the promoter of the insulin-like growth factor 2 (*IGF2*) gene that modulates foetal development and growth [27,29], and is implicated in metabolic syndrome [29]. Apart from low birth weight and CVD, early exposure to famine also caused low lipoprotein levels in offspring [30,31]. A recent study has shown an association between epigenetic changes in genes involved in the regulation of lipid metabolism and adipogenesis in offspring exposed to famine in early gestation [32]. Thus, the hypermethylated regions of the genes serine/threonine kinase Pim-3 (*PIM3* influences cell growth and energy metabolism [86]), 6-phosphofructo-2-kinase/fructose-2,6-biphosphatase-3 promoter (*PFKFB3* is involved in glycolysis), and methyltransferase 8 (*METTL8* is involved in adipogenesis) affected the expression of these genes by an increase in DNA methylation, and contributed to the development of adverse metabolic profiles. Additionally, men born after exposure to famine in early gestation had a two-fold increase in the risk of developing schizophrenia, and decreased intracortical volumes and total brain volumes, suggesting that early brain development may be a risk factor for developing schizophrenia [87,88]. 

### 3.2. Seasonal Diets: The Gambian Example

One example where seasonal diet affects offspring outcomes is in the rural economy in the African country the Gambia. This study distinguished a hungry/wet season (June–October) and a harvest/dry season (November–May). Epidemiological and longitudinal studies have suggested that there is a relationship between the timing of conception during the seasons and increased susceptibility to uterine growth restriction [89], an increase in premature death, and the development of NCD and CVD in adulthood [90].

One main objective of these studies was to find the underlying mechanisms, such as epigenetic regulation during the periconceptional period, that made these seasons change the health risk in adulthood. The modification of DNA methylation patterns has been associated with maternal diet during seasonal periods in the Gambia, since the foetal development period is sensitive to prenatal exposures, and drastic changes in DNA methylation can affect the reprogramming that occurs during embryogenesis [91,92]. Studies on Gambian populations have shown that different seasons (affecting food availability) influence the DNA methylation of metastable epialleles (MEs), since DNA methylation relies on nutritional inputs of methyl donors (folate, choline, and betaine) and essential cofactors (B2, B6, and B12 vitamins), which could be influenced by rainy and dry seasons [93]. MEs are genomic loci; DNA methylation patterns are originated from all three germ layers (before gastrulation) of MEs in an individual, and are maintained in differentiated tissues [33,93]. Therefore, the study of MEs provides a useful tool to identify the role of the periconceptional environment in the establishment of epigenetic changes, such as DNA methylation, in the offspring. Waterland et al. [33] analysed BolA family member 3 (*BOLA3*), paired box 8 (PAX8), SLIT and NTRK-like family member 1 (*SLITRK1*), and zinc finger FYVE-type containing 28 (ZFYVE28) as MEs in peripheral blood, and found that in individuals conceived during the nutritionally challenged rainy season, DNA methylation in MEs was higher. These data suggest that periconceptual nutritional status may influence DNA methylation changes at MEs in the early stages of development (preimplantation period), and impact the offspring’s health later in life [33]. Additionally, James et al. analysed one-carbon biomarkers in maternal plasma at the time of conception and 50 ME loci in their offspring’s blood in a Gambian population. They observed increased ME methylation, higher plasma folate, vitamin B2, betaine, and cysteine, and lower plasma homocysteine in rainy season conceptions. These preliminary results suggest that rainy and dry seasons could influence infant DNA methylation [34,94]. Some of these regions with altered methylations during the different seasonal periods, and their implication in the health of the offspring, have been closely related to one-carbon metabolism [93,95]. However, studies of the Gambian seasonal diet have not yet found a direct association between specific epigenetic changes and a given disease.

### 3.3. Low-Protein Diets (LPD)

Maternal malnutrition has been associated with the development of metabolic diseases in adult offspring, such as obesity, insulin resistance, and diabetes [13,17,35,96]. However, the mechanisms underlying the development of metabolic diseases in offspring have not been fully elucidated. Epigenetic modifications have been implicated as contributory since the levels of proteins and some essential amino acids and methyl group donors may influence epigenetic regulation [97]. Maternal LPD has recognisable effects from the preimplantation period through to late lactation, affecting the phenotype of the offspring, and increasing the risk of developing CVD, neurological disease, and diabetes [13,17,24,35,36,96,98,99,100,101,102]. 

In a mouse model, male offspring from mothers exposed to an LPD (10% protein) during gestation and lactation had reduced body weight compared with controls, and exhibited a loss of methyl groups from CpGs in the promoter of the leptin gene in adipose cells, coincident with reduced levels of leptin mRNA [103]. Leptin is an appetite-regulating hormone, and controls body weight through food intake and energy expenditure. Therefore, the imbalance in leptin levels, caused by maternal LPD, suggests that these alterations could not prepare the individual for an environment with an excess of food, leading to an increased risk of obesity [103,104].

Another study exploring the epigenetic changes associated with maternal LPD was conducted in a pig model [36]. The mothers were fed LPD (6%) one month before fertilisation and during pregnancy. Here, the epigenetic regulation of glucose-6-phosphatase (*G6PC*), which has a crucial role in glucose regulation and the risk of hyperglycaemia, was studied. The study showed that the *G6PC* promoter was hypomethylated in male offspring, together with increased methylation of H3K4 in the *G6PC* promoter in the liver. Therefore, LPD during pregnancy caused activation of the *G6PC* gene in males, linked with the development of hyperglycaemia and diabetes in adulthood [36]. Similar offspring outcomes were shown in another study using LPD (9% protein) in a pregnant rat model. Here, the effect of LPD on glucose transporter 4 (*GLUT4*), involved in insulin tolerance in the skeletal muscle in the offspring, was investigated [37]. Histone H3 and H4 were acetylated and histone H3K4 was dimethylated in the *GLUT4* promoter region in female offspring, providing evidence that glucose metabolism may be adapted by epigenetic processes in LPD offspring in a sex-dependent manner. The resulting low expression of *GLUT4* is associated with metabolic syndrome and the development of type II diabetes.

In a further maternal mouse LPD model, the modification of histone acetylation and increased H3K9 methylation in the promoter domain of the cholesterol 7α-hydroxylase gene (*Cyp7a1*), which catabolizes the conversion of cholesterol to bile acids, was associated with decreased expression in the liver of male offspring [105]. Moreover, similar to previous studies in mice, decreased acetylation and increased methylation of histone H3K9 were reported, which regulates the *Cyp7a1* promoter, reducing expression and increasing the risk of metabolic diseases, including insulin resistance and raised cholesterol levels, in offspring [38].

In other studies using the pig LPD (6.5% protein) model during gestation, hepatic cytochrome C gene (*CYCS)* expression was increased in the liver, and was associated with increased methylation of specific CpG sites in the *CYCS* promoter in new-born LPD offspring [40]. *CYCS* expression is associated with mitochondrial energy metabolism and production, and with increased oxidative stress. A study of ewes born on a restricted maternal diet (LPD) obtained similar results, showing increased cytochrome C protein in the liver of LPD offspring, indicating increased mitochondrial activity [41].

These varied examples of maternal LPD during gestation, across mammalian species, show how epigenetic changes, induced by certain dietary exposures, may be associated with the development of CVD, insulin resistance, eating disorders, obesity, and other metabolic diseases in offspring.

### 3.4. High-Fat Diets (HFD)

Several studies have focused on the effect of excess maternal fat and caloric consumption on offspring epigenetic changes and disease risk, such as obesity. Obesity is a complex disease caused by different factors, including high-fat diets, stress, eating disorders, a sedentary lifestyle, and genetic factors. The increase in obesity is a major threat to health worldwide, regardless of gender and age. In both developed and developing countries, a change to a more sedentary lifestyle and the excess consumption of fatty foods have had a negative effect on population health [106]. Currently, approximately 36.5% of men and 38% of women worldwide are classified as obese [107], and in the last twenty years, the prevalence of maternal obesity has increased alarmingly [108,109]. Maternal HFD (and obesity) affects the embryo and foetus during gestation, causing changes at the epigenetic level during reprogramming, promoting the propensity of the embryo to develop diseases (such as insulin resistance), due to exposure to this environment in utero [110]. Some of the epigenetic modifications due to maternal HFD will be discussed. In vivo studies are valuable in identifying the link between maternal HFD and disease outcomes in offspring, as well as in determining the most vulnerable time point in development [111]. 

During pregnancy, maternal obesity profoundly influences the risk of obesity and metabolic diseases in the offspring; for example, the offspring of mothers fed an HFD (45% fat) for 12 weeks showed a non-alcoholic steatohepatitis phenotype (most prominent in males) and altered methylation patterns (hypermethylation) in genes implicated in liver fibrosis and lipid accumulation, such as *Ephb2* (ephrin type-B receptor 2) and *Fgf21* (fibroblast growth factor 21), suggesting that epigenetic changes could favour the development of fatty liver disease in the offspring [42]. Similarly, and as mentioned above, leptin regulation is essential to prevent metabolic diseases. In a maternal HFD (60% fat) mouse model, hypermethylation at the promotor region of adiponectin (visceral fat) and leptin receptor genes (visceral fat and liver have a role in adipogenesis), and decreased DNA methylation at the promotor of the leptin gene, were reported in the adult offspring [112]. Indeed, in a mouse model, the offspring of mothers exposed to an HFD during pregnancy showed modifications in the methylation status of the adiponectin promoter region in the visceral fat, and changes in methylation in the leptin promoter region, suggesting that these epigenetic modifications may modulate adiponectin and leptin expression in the adipose tissue, potentially leading to a phenomenon similar to the metabolic syndrome [10,43]. 

Another study presented an association of epigenetic modifications in the *Pomc* (proopiomelanocortin—biomarker for obesity detection) promoters in female rats. The adult female offspring of HFD dams (60% fat) had increased body weight and blood leptin levels, and their weight gain was related to *Pomc* expression. The hypothalamic *Pomc* promoter region was hypermethylated in the offspring of HFD-fed mothers, showing a long-term effect on *Pomc* methylation programming [44]. Consistent with this notion, the offspring of female rats fed an HFD (34% fat) before and during gestation and lactation developed a metabolic syndrome-like phenotype and an altered *Pomc* promoter methylation pattern in the hypothalamus. However, the male offspring were more predisposed to alterations in the methylation patterns, and were the only offspring that showed hypermethylation in different CpG islands and the promoter region of the *Agrp* (agouti-related peptide) gene in the hypothalamus [45]. Additionally, in humans, a study demonstrated that epigenetic changes, especially leukocyte methylation levels in the promoter region of the *Pomc* gene, may be associated with weight regain in obese men [46]. The biological significance of this hypermethylation is related to the dysregulation of leptin, leptin receptors, and the Agrp-Pomc system in the brain, and weight gain in offspring, which may lead to obesity later in life. Interestingly, another maternal over-feeding model, with Wistar rats fed an HFD (34% fat) during gestation and lactation, described changes in the gene expression and hypermethylation pattern of the insulin receptor (InsR) DNA promoter in the hypothalamus in a sex-specific manner, leading to a metabolic syndrome-like phenotype in the male offspring; however, lower methylation levels of *Pomc* were associated with weight loss [46].

In a macaque maternal HFD (35% fat) model during gestation, the offspring exhibited hyperacetylation of the foetal hepatic tissue at H3K14, and increased H3K9 and H3K18 acetylation (not significant) [47]. These histone changes were related to an increase in the expression of offspring genes, including glutamic pyruvic transaminase 2 (*GPT2*, which is essential in the maintenance of obesity in the postnatal period) and retinol dehydrogenase 12 (*Rdh12* regulates feeding behaviour and adaptation to circadian rhythms). *Rdh12* is also related to the family of short-chain alcohol dehydrogenases [48], so its deregulation could cause non-alcoholic fatty liver disease [49]. Additional findings in the foetal heart of non-human primate offspring, following maternal HFD and obesity during pregnancy, showed dysregulation in 80 miRNAs [51] involved in the migration of the cardiac neural crest during development [113].

A recent study using HFD (60%) during pregnancy and lactation in rats found that maternal HFD affects cannabinoid receptor 1 (*Cnr1*) expression and its underlying epigenetic status in the brain of offspring at early ages [20]. Gawlinski et al. found that *Crn1* mRNA decreased and miR-212-5p and CpG island methylation increased in the *CnIr1* promoter in the cortex of male offspring before adulthood. Interestingly, these authors found the opposite effect in the hippocampus of adolescent male offspring; the mRNA of the *Cnr1* increased, thus increasing its expression and, at the same time, decreasing the expression of miR-212-5p and the methylation of CpG islands in the gene promoter. The authors related these expression changes in the *Cnr1* to the coding of the CB1 receptor (related to depressive behaviour), and to the phenotype observed in the offspring, who presented a depressive-like phenotype [20].

These examples show that maternal HFD during gestation produces changes in the offspring epigenome across different mechanisms and species, including primates, which increase the risk of developing diseases such as non-alcoholic fatty liver disease, diabetes, obesity, CVD, and other NCDs.

### 3.5. One-Carbon Metabolism: Folic Acid Supplement Use 

One-carbon metabolism is related to pathways involved in the methylation of proteins, nucleic acids, and phospholipids, such as the pathway of folate and methionine, among others. The nutrients associated with one-carbon metabolism (riboflavin, folates, B12, among others) must be obtained from food because the body does not synthesize them. However, these nutrients are very important for the development of the gametes, embryos, and the health of the offspring later in life [114,115,116]. Alterations in the uterine environment, such as changes in the maternal diet, can compromise one-carbon metabolism regulation, which can trigger endocrine and metabolic changes during development [117,118]. One-carbon metabolism covers a large number of metabolic pathways that supply methyl groups required for DNA methylation [119]. Methylation reactions are imparted during embryonic and foetal development; therefore, dietary deficiencies of certain nutrients associated with one-carbon metabolism (i.e., vitamin B6, vitamin C, folate, choline, etc.) can affect epigenetic processes in utero, leading to changes in methylation that may promote the development of diseases in the offspring later in life [120,121]. In this review, we focus our one-carbon metabolism discussion on the importance of folate in the maternal diet during the preimplantation period and gestation. One of the best-known examples is the absence of folates in the maternal diet during the first months of pregnancy, which can cause diseases during neural tube development in the foetus (“spina bifida”) [122,123]. Other studies in human cohorts found an association between folic acid supplementation during the periconception period and increased methylation in *IGF2* in the offspring [124]. DNA methylation during development is related to methyl donor groups obtained from one-carbon metabolism, so there are several associated studies [115,119,125,126,127,128,129,130].

In vivo studies of a rat diet comprising a maternal folic acid supplement used during pregnancy showed that nutrient imbalance produced global DNA hypermethylation in the cortex of the adult offspring [52]. These animals were analysed in behavioural tests, and a longer escape latency was observed compared to the controls. The cortex is associated with cognitive behaviour, decision making, and memory, so changes in the DNA methylation of this brain region can trigger brain disorders, such as Alzheimer’s disease and bipolar disorder [53]. Folic acid supplement use (5 mg/day) during pregnancy in rats induced a significant reduction in DNA methylation in the mammary glands of adult offspring [54]. Surprisingly, these findings have been associated with an increased risk of breast cancer, so high folate levels during pregnancy and lactation are associated with an increased risk of breast cancer in offspring. 

A mouse embryonic stem cell culture model has been used to determine the effects of restricted folate supplementation (0.5 mg/L folate) in early development, revealing hypomethylation of the long interspersed nuclear element-1 (LINE-1) promoter domain and increased expression of LINE-1, which is related to neural tube defects [55]. This change was associated with decreased transformation of stem cells into embryonic neural tissue, indicating neural tube defects. 

The above examples of manipulating folic acid supplement use during pregnancy elucidate some of the epigenetic mechanisms that are regulated. An unbalanced folate diet can induce the development of neurological diseases, and even cancer, in offspring. However, more studies on the underlying mechanisms are needed to better understand folic acid supplementation.

A summary of all the examples revised in the different diet models mentioned above is shown in Table 1. 

## 4. From Preimplantation Period to Development: Epigenetics and Thrifty Genes

Nutrient accessibility during pregnancy affects crucial events, such as cell growth and division, that are necessary for optimal gametogenesis and embryo development [131,132]. One of the periods on which we would like to place special emphasis, and in accordance with the premises of the DOHaD hypothesis, is the preimplantation period. The preimplantation period encompasses all the morphological, structural, and epigenetic (mainly DNA methylation and histone modifications) enhancements and remodelling that occur in the embryo until just before implantation in the uterus [133]. This is a very short period, but is highly vulnerable to changes in the environment, in which epigenetic modifications can affect gene expression in the developing embryo, which can lead to health defects in adult life; for example, it has been shown that dietary changes during the preimplantation period alone (especially in murine models) can lead to neurological pathologies, NCD, and CVD in the offspring during adulthood [13,131,134,135]. One possible explanation for these events is the thrifty gene hypothesis. This hypothesis was proposed by the geneticist James Neel in 1962, its premise being the study of diabetes and the genes that promote its expression [136]. This hypothesis refers to hunter-gatherer populations, in which the predisposition of these genes allowed them to accumulate fat more efficiently during periods of greater nutrient abundance. Thus, the hypothesis proposes that the expression of certain genes, whilst increasing disease susceptibility in later life, may have, historically, been advantageous in periods of famine or lack of nutrients, but today, the expression of these gene may be detrimental [136,137,138]. Over the years, some authors have proposed some candidate thrifty genes. However, we have found that few have linked these changes to maternal diet [139]. 

One of the basic units of cellular protein translation and cellular growth is the ribosome, and it has been shown that rDNA can be activated by epigenetic mechanisms [140,141]; for instance, the transcription of rDNA can be silenced by hypermethylation of the rDNA promoter, induced by environmental conditions [140]. Through hypermethylation of the rDNA promoter, rDNA transcription can be silenced, so if the environmental conditions to which the mother is exposed can affect epigenetic regulation, they may also affect rDNA expression for development [142]. Ribosomal RNA expression is sensitive to maternal diet, and, using the mouse LPD model, it was shown that poor diet during the preimplantation period was sufficient to activate a lifetime pattern of enhanced rRNA expression, only when the dietary restriction was lifted during either the foetal or adult stages, reflecting a thrifty phenotype response [142]. A similar effect in response to gestational HFD on epigenetic changes in rRNA expression has been reported [143]. A more recent study has proposed the phosphatase and tensin Homolog (PTEN) as a possible thrifty gene that undergoes epigenetic modifications related to maternal diet. However, the epigenetic regulations involved that affect PTEN expression/silencing have not yet been found [139]. 

Therefore, we consider that the study and identification of affected genes by maternal diet models could be of great interest Thus, epigenetic changes that can occur in these genes during the preimplantation period can lead to the phenotype of the offspring being altered.

## 5. ART and Epigenetic Modifications

ART includes a range of techniques, such as superovulation, sperm capacitation, intrauterine insemination (IUI), intracytoplasmic sperm injection (ICSI), in vitro fertilization (IVF) and culture, cryopreservation, and embryo transfer for infertility treatment in humans and the production of transgenic farm animals [144,145]. ART treatment can interfere with epigenetic changes, particularly DNA methylation [146], starting from the preimplantation period in humans and animals [147,148], affecting early embryogenesis and offspring health [149,150,151]. The procedures implemented during ART treatment, such as ovarian stimulation using hormones, the in vitro maturation of oocytes and preimplantation embryos, and the use of ICSI and embryo cryopreservation, expose the embryo to environmental conditions that differ from those in spontaneous fertilization and embryogenesis, and which may alter the normal epigenetic mechanisms [149,152].

### 5.1. Human Studies

DNA methylation, following ART, has been investigated at different stages of development. Genome-wide DNA methylation profiling in cord blood and placenta revealed differences in the methylation status between ART and spontaneous conceiving populations at certain CpG sites [153,154,155,156]. In ICSI-manipulated oocytes, this can include an abnormal methylation status at imprinted genes at differentially methylated region 1 (DMR1), H19 DMR, and PEG1 DMR [157]. Following ICSI treatment, day 3 embryos also showed aberrant methylation patterns at the same imprinted loci [158]. The aberrant methylation pattern of these DMRs is linked to imprinting disorders, such as Silver–Russell syndrome (SRS) and Beckwith–Weidemann syndrome (BWS) [159,160,161,162,163]. All the BWS patients conceived via ART had DNA methylation errors at H19/Igf2 IG DMR and/or KCNQ10T1:TSS DMR, and the DNA methylation error rates were significantly higher when compared to the spontaneously conceived BWS patients (100% versus 43.6% DNA methylation error rates, respectively) [151]. Interestingly, the DNA methylation levels of H19/Igf2 DMR and KCNQ10T1 DMR were significantly lower in the placentas of IVF/ICSI patients compared to the spontaneously conceived patients [164]. However, no differences in DNA methylation levels were reported between IVF and the spontaneous controls at individual CpG sites and entire DMRs of KvDMR1 in cord blood and placental villi, as well as PEG10 in placental villi [165]. These findings suggest that certain epigenetic variations in ART-conceived embryos may resolve throughout development, whereas others do not, and might associate with offspring outcomes. To further support this hypothesis, a positive correlation was observed between the DNA methylation of ERVFRD-1 and placental weight, as well as ERVFRD-1 and SNURF with birth weight, whereas the epigenetic variations at birth, as a consequence of ART, were largely resolved by adulthood [164,166]. Moreover, the epigenetic changes at certain genes and the transmission to offspring might be tissue-specific [164,165]; for example, the differences in methylation levels between ART and spontaneous conceptions for the H19/Igf2 and KCNQ1OT1 DMRs observed in the placenta were absent in the cord blood [164].

### 5.2. Animal Studies

Animal studies provide more insight into the prenatal period to help understand the origin and flow of epigenetic changes. The profiling of DNA methylation levels in mouse embryonic cells, blastocysts, the placenta, and the foetus displayed differences between ART and the controls [152,167,168,169]. Similarly to humans, some epigenetic alterations at certain genes among the ART population arise during the periconception and preimplantation period, and can be maintained throughout pregnancy. Aberrant methylation patterns at certain CpG sites of the Igf2/H19 imprinting control region (ICR) are observed in the IVF mouse embryonic stem cells (ESCs), placenta, foetal brain, and liver tissues of certain mouse strains [170,171]. Differences in methylation status at the H19 ICR were also reported in ART blastocysts, embryos, aorta, and placental tissues [149,172]. Nevertheless, the presence of alterations might also differ among the crossed strains [150,171,173]. A reduction in methylation for KvDMR1 and *KCNQ1OT1* was observed in the ART placentas compared to the controls, whereas no difference was observed in the foetal brain and liver tissues [150,152,171,172]. Other examples are the small nuclear ribonucleoprotein polypeptide N *(Snrpn),* PEG1 and PEG3 ICRs that showed an altered methylation status in certain tissues, whereas no difference was observed in others [150,171,172,174]. Altogether, these alterations can be correlated to the aberrant expression levels of the relative genes, blastocyst maturity, and foetal development, and the imprinting disorders observed among ART offspring [149,152]. Moreover, the alterations in the methylation levels of SCAP/SCREPF1-2 in the lungs and the promoter of the endothelial nitric oxide synthase (eNOS) gene in the aortic and vascular tissue implicate the role of epigenetic alterations as a basis for the cardiovascular and respiratory dysfunction observed among ART-conceived offspring [167,174,175,176].

Whether these epigenetic changes are due to the underlying subfertility or to the ART treatment in humans warrants further investigation, yet similar datasets from animal models suggest that the effect of ART cannot be ignored; for example, ART was associated with alterations in DNA methylation in embryos at H19 and PEG 1 DMRs in both human and animal models [158,172]. In placentas, an alteration in the *KCNQ1OT1* methylation status was associated with ART in human and animal models [164,171], whereas the outcomes on KvDMR1 and PEG10 ICR were inconsistent between humans and animals [150,165]. Besides, differences in methylation patterns have also been reported between different ART procedures [149,152,164,168], which emphasizes the effect of ART manipulation on the epigenetic profile and subsequent disease risk 

## 6. Diet and ART Outcomes

In the sub-fertile population, diet has been related to fertility [177,178] and implicated in the intermediate ART outcomes, such as fertilization rate [179] and embryo yield [180], as well as clinical outcomes, including, importantly, pregnancy [181] and live birth rates [179]. Adherence to a healthy dietary pattern increased the number of mature oocytes [181,182]. Higher fertility and implantation rates were associated with the Iranian traditional medicine (ITM)-based diet and “pro-fertility” diet, respectively [181,183]. Increased embryo yield was observed with higher adherence towards the Mediterranean and ITM-based diets [180,181]. The chance of pregnancy was also improved with increased adherence to the “pro-fertility”, Mediterranean and ITM-based diets [181,183,184,185]. Furthermore, Twigt et al. reported an association between the chance of ongoing pregnancy and higher preconception dietary risk scores [186]. Besides, the rate of live birth was also improved by healthy dietary behaviours in women [179,183,187]. In contrast, unhealthy dietary behaviour was linked to a lower mature oocyte count, embryo quality, pregnancy chance, and live birth rate in human and mouse models [182,188,189]. Moreover, Li et al. investigated the effect of antioxidant intake, and showed improvements in the oocyte yield and live birth rate, which were dependent on the source and type of antioxidant [190].

A diet–epigenetic–ART connection might, in part, explain the ART outcomes observed, yet studies investigating this connection are limited [172,191]; for example, a healthy and balanced maternal diet could, through correcting the epigenetic alterations caused by ART, modify the corresponding outcomes [172,192]. On the other hand, an unbalanced diet (i.e., HFD diets) may pose epigenetic alterations that might not be observed otherwise [193]. A suggested mechanism behind this connection can be related to the effect of diet on the molecular interactions and biomarker levels within the body, which can induce epigenetic alterations; for example, improved pregnancy and live birth rates were associated with folate intake, which was also observed with adherence to the Mediterranean diet, which showed an association with the folate and vitamin B6 levels [179,184,185]. Subsequently, Rahimi et al. showed that DNA methylation defects induced by ART were partially restored with moderate folic acid supplementation in a mouse embryo and placenta [172]. In terms of phenotypic changes, moderate folic acid supplementation reduced the number of embryos with developmental delays, associated with ART treatment [172], which can further affect the disease risk in offspring [194,195].

## 7. Conclusions

Maternal diet could induce metabolic and physiological changes in offspring through epigenetic modifications. The altered epigenetic regulation of genes is associated with increased predisposition to disease later in life. Despite the advances in sequencing, there are still many mechanisms that need to be studied to understand the exact contributions of developmentally induced epigenetic markers and their effect on the risk of disease development. Understanding the relationship between maternal diet during pregnancy, epigenetic markers, and disease development may allow the discovery of therapeutic targets for the prevention and treatment of NCDs.

There is now great interest in understanding the environmental basis affecting epigenetic modifications that may be the origin of diseases in offspring. In humans, we have mentioned examples of epidemiological and ART-associated studies that could be used to understand the origins of some diseases or syndromes. However, animal studies are more flexible and controlled, as defined conditions can provide us with more information on the environmental effect during embryonic development and the outcome of the offspring in a shorter time, and, interestingly, we found that in both the natural condition and ART, maternal diet influences the offspring outcomes (Figure 1).

Moreover, the theory of thrifty genes provides an interesting area to understand the epigenetic regulation during the preimplantation period. For this reason, we consider that the study of these genes during early development can be crucial to understand the effect of maternal diet on epigenetic modifications.

## Figures and Tables

**Figure 1 genes-13-00031-f001:**
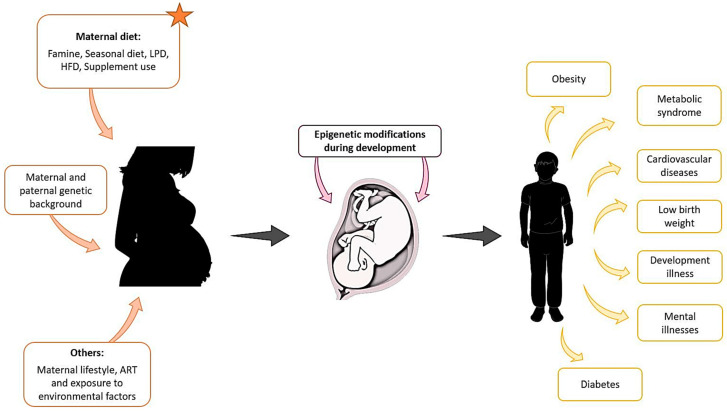
Maternal exposure to environmental factors, diet changes, genetic background, and other parameters such as lifestyle, can affect the development of the foetus from the first stages of pregnancy and compromise the health of the offspring later in life.

**Table 1 genes-13-00031-t001:** Maternal diets models and epigenetic modifications in offspring.

Maternal Diet	Offspring Effect	Epigenetic Modification	Species	Observed Outcome	References
Dutch famine (Undernutrition)	↓methylation IGF2	DNAmet	Human	Low birth weight, CVD and low lipoprotein levels	[27,29,30,31]
Dutch famine (Undernutrition)	↑methylation PIM3 ↑methylation PFKFB3 ↑methylation METTL8	DNAmet	Human	Increased risk of metabolic diseases	[32]
Seasonal diets (The Gambian example)	↑methylation in MEs when conception during hungry season	DNAmet	Humans	Have not yet found a direct association of specific epigenetic changes, but there is an increase in disease risk	[33,34]
LPD (6% protein)	↓methylation G6PC	DNAmet	Pig	Hyperglycaemia in adulthood	[35,36]
LPD (9% protein)	H3, H4 acetylated ↑methylation H3k4 ↓GLUT4 expression	Histone acetylation and methylation	Rat	Insulin tolerance Metabolic syndrome	[36,37]
LPD (8% protein)	↑methylation H3K9 ↓Cyp7a1	Histone methylation	Rat	Metabolic diseases, long term increase in cholesterol	[37,38,39]
LPD (9% protein)	H3, H4 acetylated ↑methylation H3K4 ↓GLUT4 expression	Histone acetylation and methylation	Rat	Insulin tolerance Metabolic syndrome	[37,38,39]
LPD (6.5% protein)	↑methylation CYCS promotor	DNAmet	Pig	Mitochondrial energy metabolism and production	[37,40,41]
HFD (45% fat)	↑methylationin *Ephb2* and *Fgf21*	DNAmet	Mouse	Non-alcoholic steatohepatitis phenotype (most prominent in males)	[42]
HFD (60% fat)	↑methylation Leptin promotor	DNAmet	Mouse	Insulin resistance	[10,40,41,43]
HFD (60% fat)	↑methylation Pomc promotor	DNAmet	Rat	Eating disorders, insulin resistance	[10,43,44]
HFD (34% fat)	↑methylation Pomc promotor in the hypothalamus	DNAmet	Rat	Metabolic syndrome	[45]
HFD (34% fat)	↑methylation InsR in the hypothalamus ↓methylation Pomc promotor	DNAmet	Rat	Metabolic syndromeWeight loss	[46]
HFD (35% fat)	Hyperacetylation H3K14, H3K9 and H3k18 on the promotors of GPT2 and RDH12	Histone acetylation	Macaque	Obesity and non-alcoholic fatty acid liver disease	[10,47,48,49,50]
HFD (maternal obesity)	downregulation in miR-181a	Dysregulation miRNA	Primate	CVD and heart development	[47,48,49,51]
HFD (60% fat)	Cortex: ↓mRNA Crn1, ↓ mimiR-212-5p and ↑ methylation Crn1 promotor. Hippocampus: ↑ mRNA Crn1, ↓ mimiR-212-5p and methylation Crn1 promotor	Dysregulation miRNA	Rat	Depression-like behaviour	[20,51]
Folic acid supplement use	↑methylation -Hypermethylation in cortex	DNAmet	Rat	Brain disorders	[20,52,53]
Folic acid supplement use (5 mg/day)	↑methylation mammary glands	DNAmet	Rat	Breast cancer	[52,53,54]
Low folic acid supplement use (0.5 mg/day)	↓methylation -hypomethylation in the LINE-1 promoter	DNAmet	Mouse embryonic stem cells	Neural tube defects	[54,55]

Abbreviations: DNA methylation (DNAmet), low-protein diet (LPD), high-fat diet (HFD). Significant increases (↑) or decreases (↓) in the epigenetic modification of the genes mentioned are represented by an arrow.

## Data Availability

Not applicable.

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
