# Peer review of "Epigenetics in the Uterine Environment: How Maternal Diet and ART May Influence the Epigenome in the Offspring with Long-Term Health Consequences"

_genes, 2021, doi:10.3390/genes13010031_

Round 1

Reviewer 1 Report

This manuscript is of potential interest and importance, however it requires extensive editing before it can be considered for publication in the Journal. Some specific points are listed below:

Line 37: It says “DOHaD … has demonstrated”. Hypothesis cannot demonstrate anything

Line 87: “In early pregnancy periods, maternal malnutrition can alter the epigenetic pattern of … gametes” hinder

Line 89: “The intrauterine environment is susceptible to any small changes, …” does it mean that this environment is sensitive to small changes in diet (and other exposures), or do you want to say that it varies all time (independent of exposures)?

Line 92: Who has shown that “genotype of the next generation” can be affected by maternal diet? It’s a strong statement, please provide a reference (ref 19 is a review, not the original work).

Line 96: “… how lifestyle and dietary behaviours can trigger changes in the DNA”. What changes are you referring to? It cannot be about epigenetics (because you said before that epigenetics does not involve changes in DNA), can it?

Line 98: “However, these changes may not occur randomly as a result of dietary variation”. This statement is confusing. Do you claim that there are random changes in gene expression induced by dietary variation? Please give an example

Line 103: The whole chapter should be extensively edited by a person with expertise in DNA methylation. The same should be done with the histone modification chapter

Line 104: “DNA methylation is the binding of a methyl group to the DNA”. Sounds like methyl groups present on any molecule can bind to DNA, which is wrong. DNA methylation is not “binding of a methyl group to the DNA” but attachment of methyl groups to cytosine (to its 5th carbon, not “fifth cytosine position”). Delete this sentence

Line 107: “The CpG sites in the genome are often methylated, which in general causes gene expression to be reduced, however, if the CpG sites are not methylated…” It’s a confusing and generally wrong way to put. First, what do you mean by “often”? Do you want to say “dynamic” or “large fraction of CpGs is methylated”? Second, well expressed genes are highly methylated within the gene body. You must be referring to promoter methylation which is inhibitory? Be more specific or delete this sentence.

Line 111: Reference 121 is wrong, by no means these authors were the first who have shown that

Line 112: is it correct to say “fertilization has been established”? “nonimprinted genes” cannot be “DNA methylation markers”. The whole sentence is wrong, delete it or say “DNA methylation is erased at nonimprinted genes”

Line 114: Followed, not “flowed”

Line 115: Explain why is it “interesting”

Line 117: How can methylation be maintained by mitotic cleavages?!

Line 117: Why is it “also” essential?

Line 117: What do you mean by “establishment of new tissues”?! Do you want to say “tissue differentiation” instead? If there are “new tissues”, there must be “old tissues”. Do you claim that DNA methylation is essential only for the establishment of new tissues, but not for maintenance of differentiated state? It is wrong

Line 118: References 14 & 22, these authors were not first to show that DNMTs methylate DNA

Line 120: Reference 14 is wrong, use appropriate ones

Line 120: What do you mean by saying “however”?! Delete it

Line 122: Delete “For this reason”; delete “chromatin”

Line 126: “the embryo may undergo DNA methylation in utero…” Do you want to say that some embryos may while others may not undergo DNA methylation in utero?

Line 127: “DNA methylation … is directly influenced by exposure to these substances”. DO you want to say that arsenic, for example, directly (without any other factor) methylates of demethylates DNA? It is not possible, is it?

Line 132: One modification?

Line 133: Delete “but”

Line 134: “The terminal tail domain” – Which one?

Line 136: “… opening or closing the chromatin state” - How can one open (close) state of something? It can be done with chromatin structure, not state.

Line 139: “… depending on the regions where the modified gene is located” - What do you mean by that? Explain (and provide references) or delete this part.

Line 140: “the residues” – substitute with “histone amino acid residues”

Line 140: References 28 and 29 do not represent histone modification field of research.

Line 145: “less than 2% of the 145 genome encodes proteins” – Which genome are you referring to? There is no such thing as “eukaryotic genome”  

Line 145: Provide a reference about 70%/2%

Line 146: “Therefore, there is a large part of the genome that does not encode proteins, known as noncoding RNA” – It is meaningless, no one part of any genome is known as RNA (except some viruses)

Line 150: “can alter functions” – What functions?

Line 152: “thus, their main function is the inhibition DNA expression” – What do you mean by that?

Line 175: “A recent study has shown epigenetic changes in the regulation of lipid metabolism and adipogenesis” – Epigenetic changes regulate genes, not lipid metabolism. Was this study done in animals or humans?

Line 195: “… and the risk of disease has been DNA methylation” – How can DNA methylation be “the risk of disease”?!

Line 203: “… preimplantational environment has a permanent effect in adulthood” – What is it about?

Line 206: “studies of the Gambian example” – How can one study an “example”

Line 212: “Proteins are essential not just for growth but also contribute to uterine-embryo communication for implantation and in DNA methylation during development” – This sentence is meaningless.

Line 222: What does “reduce … body composition” mean?

Line 300: How can hyperacetylation of H3K9 cause “reduction in adiponectin expression”? Isn’t it opposite?

Line 302: “these variations are related to leptin and insulin resistance in adulthood, which may result in the development of the obesity phenotype…” – has it been shown that these changes last to adulthood?

Line 355: It is “nuclear”, not “nucleotide”

Table 1: In position 3 (seasonal diets), what was the Observed outcome (disease risks or else)?

Table 1: After the table, briefly summarize data and tell us what genes are common to these models, if any. What can you say if there no genes common to these models?

Line 394: Explain what is “100% DNA methylation errors”

Line 395: “… and were significantly higher” – What was higher (compared to what?)?

Line 403: “specific transposable element and DMR” – Define the element and the DMR

Line 406: “Moreover, the epigenetic change at certain genes and transmission to offspring might be source-dependent” - Do you want to say the epigenetic changes are tissue-specific?

Lines 433-438: This paragraph is hard to understand. In the end it says: “the importance of ART manipulation on epigenetic profile and disease risk”, which suggests that epigenetic profile and disease risk are independent of each other. Is it your message? I would substitute this paragraph with discussion of similarities and differences in epigenetics of ART in human and animals.

Line 457: Were phenotypic changes (in disease risks) also induced by the folic supplement?

Line 469: How relevant is gametogenesis during pregnancy to the topic of this review?

Line 470: “one of the main targets of the DOHaD hypothesis focuses on…” – How can “target of a hypothesis” focus on something?

Line 486: “but today those genes may be detrimental” – How did we (humans) menage to survive with such detrimental genes? Did you want to say “gene expression” instead of “gene” here?

Line 487: “However, we have found that few have linked these changes to maternal diet” – give a reference

Line 493: “it was shown that rDNA transcription rates change mediated by rDNA methylation” – Did you want to say that transcription changes were mediated by changes in DNA methylation? Give more details about these changes

Line 494: “Pol I transcription factor” – Which one?

Line 494-496: This sentence is not clear

Line 510: So, in humans and animals, poor diet during pregnancy similarly causes poor health outcomes. Did those molecular studies reveal genes (and epigenetic changes) common to humans and animals? If not, why?

Reviewer 2 Report

The review topic is very relevant. The impact of maternal nutrition on the increased risk of NCDs in adult offspring is beyond doubt. The role of mediation of these effects by epigenetic mechanisms is a burning issue. The authors aim to discuss the evidence of how maternal diet can induce epigenetic modifications in offspring leading to the development of disease in adulthood. After an introduction, the authors present results of different studies, epidemiological and on animal models. Then, the authors explored impact of maternal diet on ART outcomes.

Plan and aims of each part are difficult to follow. Descriptions of studies should be more precise, more comprehensive. The authors discuss several very interesting studies showing epigenetics defects in offspring of under/over/mal-nourished mothers. But the authors should be careful not to present results showing associations or correlations as causal results. The interpretations and conclusions must be globally moderate. Many misinterpretations and many inappropriate references.

Introduction is quite long, and confused.

L501-51: Distinction of extrinsic and intrinsic factors is difficult to understand.

L54 “These demonstrate that the origin of NCDs was attributable to maternal nutrition and gestational environment laying the foundation for DOHaD”. The authors should moderate this sentence. Origins of NCDs are multiple and not only “attributable to” maternal nutrition and gestational environment.

L61 “These changes act to coordinate the physiological, metabolic and growth characteristics and are mediated through epigenetic mechanisms [9,10]”. The use of “mediation through” should be moderate. Mediation through epigenetic mechanism are suggested. This moderation is well explained in the Yamada et al reference (11): “Finally, it is often difficult to discern whether epigenetic marks, such as DNA methylation and post-translational histone modifications which are associated with gene expression changes following an early-life environmental exposure, are a cause or a consequence of the change in transcription”.

L87-89 “In early pregnancy periods, maternal malnutrition (overnutrition or undernutrition), can alter the epigenetic pattern of mother, gametes and embryo genomes”. But in the associated ref (17), Fleming et al. did not explored epigenetic mechanisms.

L92-95 “Different studies pointed to the possible idea that across generations, a compromised diet can affect the phenotype and genotype of the next generation, making them prone to develop metabolic and even mental illnesses, these changes may be caused by epigenetic variations which are subject to variations in diet [19]”. Ref 19, completely inappropriate, explored direct effects of nutrients on brain function, but not in a DOHAD context.

The second section summarizes the main types of epigenetic regulation of gene expression that occur during development. “Maternal diet, epigenetics and DOHaD” title of this section is inappropriate. This section should be focused on a brief description of epigenetic mechanisms. For that purpose, the first sub-section (2.1 methylation) should be rewritten to be more focused. For example L124 to 131 (Some studies have … regions involved have not yet been determined) are off topic. The next sub-section (2.2 Histone modification) should be more complete, more precise and especially referenced. This 2.2 section lacks of appropriate ref. The 2.3 subsection (Non coding RNAs) should be rewritten to be more consistent. The mechanisms of action of lnc RNAs should be explained.

In the third part (3. Diet models and epigenetic modifications: examples and biological meaning), examples are globally well chosen but their descriptions are imprecise and confusing.

Concerning 3.1 Dutch famine, it is important to highlight that in addition to CVD risk, the risk of mental illness is also increased in the offspring. The authors should also detail sexual dimorphism, in this and other sub-sections.

-L169-170 “Moreover, limited nutrient intake in early pregnancy decreased the development of pancreatic beta cells resulting in glucose intolerance later in life [36,37]”. In the ref 37, the authors have explored placental defects and not the defects in pancreas.

L 175-176 “A recent study has shown epigenetic changes in the regulation of lipid metabolism and adipogenesis in offspring exposed to famine in early gestation [34]”. In the cited ref, the authors exhibited associations but not causal link. This sentence should be rewritten.

Regarding on 3.2 Gambian example, the explanations on ME are confusing and should be rewritten. The conclusion on the impact of the preimplantation embryo in this context should be deleted because they are not supported by the previous explanations. Cited ref should need to be better understood. The implication of SLITRK1 and PAX8 are off topic.

Regarding on the 3.3 LPD, chosen studies on LPD in pig and mouse are interesting. But the advantages of mouse model are not well described. In animal model, key period could be explored. Impact of diet on gametes vs preimplantation period vs gestation vs lactation. Each period can be explored separately. This section lacks of crucial references on mouse LPD that target several developmental key periods like the embLPD model. Sexual dimorphism should be mentioned. This section should be rewritten to be more complete.

L223-227: LPD induced increased DNA methylation in the Leptin promoter in adipose cells during gestation, but from whom? Mother? Offspring? These sentences should be rewritten.

L260-263 “predispose” should be moderate.

The 3.4 HFD part has the same imprecisions.

L273-276, “Maternal HFD affects the embryo… causing changes at the epigenetic levels”. This sentence is not justified by reference, by example.

L281-288, leptin methylation results are not clear, not well described, lacks of precision. From which? Sex? Age? Tissues/cells?...

L289-294, Some evidence of causal effect of POMC promoter hypermethylation?

Table 1 should be referenced earlier in the text. Period of diet, sex and age of offspring should be added.

The aim of the last section (4. Diet and ART) is difficult to understand. It is one thing to describe the evidence of ART induced epigenetic changes (4.1 and 4.2) and another to explore impact of diet on ART outcomes (4.3). Authors intentions are not clear, and sub-section 4.3 should be rewritten to be more understandable and comprehensive.

Reviewer 3 Report

The paper submitted by Peral-Sanchez and collaborators is very interesting and concerns the information concerning the effect of maternal diet on epigenetic changes in offspring. This review is well organized and very useful for readers in this area. However, I have some minor issues how to improve this work:

1. Abbreviations should be defined the first time they appear, when defined for the first time, the abbreviation should be added in parentheses after the written-out form, for example, lines: 22, 144, 211, 264, 366-367, 429, 502, etc. Please check the whole text. 

2. To summarize the content, one or two Figures in color should be presented to facilitate following the conclusions and to increase the readability of the work.

3. Footnote for the Table 1 should be added, which contains all abbreviations and characters explained 

4. The effect of maternal high-fat diet should be enriched for their effect on depression-like behavior in offspring, especially 
Gawliński et al., Nutrients, 2021. Maternal High-Fat Diet Modulates Cnr1 Gene Expression in Male Rat Offspring. doi: 10.3390/nu13082885, which showed how maternal high-fat diet alters gene expression, miRNA, and methylation of CpG islands in brain structures.

Round 2

Reviewer 1 Report

The manuscript reads better, yet some issues still need to be corrected. Most corrections are needed in the first half of the manuscript. It would be very beneficial for the manuscript if authors who wrote and edited the second half of the manuscript (from chapter 3.2 it reads well) also help to edit/correct it’s first half.

Specific points:

Introduction to epigenetics (chapter 2) should be reduced to one-two paragraphs with brief description of major modifications, cognate enzymes and main function in gene expression. Currently, description is very redundant.

Line 59: Authors call pollution, exposure to chemicals etc. “extrinsic factors”, which is correct. However, they define “extrinsic” factors as those that are difficult to modify, which is wrong. One can easily escape/decrease exposures to chemicals (pollutants, etc.) by taking a plane (bike, horse, etc.) and go to a safer place. Instead, extrinsic factors are those that act from outside, while intrinsic act from within an individual.

Line 75: “Epigenetic changes … can be altered”, - did you want to say “epigenetic pathways … can be altered?

Line 78: “These epigenetic modifications … could be either a cause or a consequence of changes in the environment” - epigenetic modifications cannot cause changes in the environment!

Line 96: “We will discuss different maternal diet…” - word “diet” has already been used in the previous two sentences, what do authors mean by “different maternal diet”?

Line 125: Can you one or two specific examples of DNA methylation changes triggered by maternal exposure that persist to adulthood? At least refer to your own Table 1

Line 135: This sentence and the next are redundant

Line 157: “…associated with changes in DNA methylation” - what changes are you referring to, in the mother or in the fetus?

Line 161: “irregular changes in DNA methylation” - did you want to say “aberrant changes”? I do not understand what “irregular changes” means.

Line 166: Give several specific examples of histone modification changes induced by intrauterine exposures, or refer to your Table 1

Line 178: “depending on the residues the regions where the modified gene is located” - no clue what it means

Line 179: “In the histone tail, methylation of lysine (K) residues can occur, which depending on the lysine involved will be either a repressive or an active marker” - so poorly written! What “tail” (there are two “tails” in each protein)? What do you mean by “methylation of lysine (K) residues can occur”? Do you know other residues that are also methylated? Ask someone to read it and give you feedback

Line 189: Why would you discuss small RNAs here if you say in the end of this section that they are not relevant to maternal programming? Why don’t you discuss other topics that are also not relevant to DOHAD?

Line 197: “sncRNA when they are in a promoter region” - no clue what is means, and it sounds very redundant with the following sentences

Line 201: This sentence is so poorly written. And the following sentences in this paragraph also need extensive editing, can someone read it and give you feedback?

Reviewer 2 Report

The authors have successfully answered all questions posed. 

Author Response

Dear Revier 2, 

Thank you for reviewing our manuscript. We appreciate the time you spend evaluating this manuscript review again. 

Thank you for your appreciation of the comments that have been successfully answered. We hope the manuscript can now meet the journal publication requirements.

If there should be any further questions, please do not hesitate to contact me.

Yours Sincerely,

Irene Peral-Sanchez,

First author